# Hyaluronic Acid-Conjugated Carbon Nanomaterials for Enhanced Tumour Targeting Ability

**DOI:** 10.3390/molecules27010048

**Published:** 2021-12-22

**Authors:** Oisin Kearns, Adalberto Camisasca, Silvia Giordani

**Affiliations:** School of Chemical Sciences, Dublin City University, Glasnevin, D09 E432 Dublin, Ireland; oisin.kearns2@mail.dcu.ie (O.K.); adalberto.camisasca@dcu.ie (A.C.)

**Keywords:** hyaluronic acid, carbon nanomaterial, carbon nanotube, graphene, graphene oxide, graphene quantum dot, drug delivery

## Abstract

Hyaluronic acid (HA) has been implemented for chemo and photothermal therapy to target tumour cells overexpressing the CD44^+^ receptor. HA-targeting hybrid systems allows carbon nanomaterial (CNM) carriers to efficiently deliver anticancer drugs, such as doxorubicin and gemcitabine, to the tumour sites. Carbon nanotubes (CNTs), graphene, graphene oxide (GO), and graphene quantum dots (GQDs) are grouped for a detailed review of the novel nanocomposites for cancer therapy. Some CNMs proved to be more successful than others in terms of stability and effectiveness at removing relative tumour volume. While the literature has been focused primarily on the CNTs and GO, other CNMs such as carbon nano-onions (CNOs) proved quite promising for targeted drug delivery using HA. Near-infrared laser photoablation is also reviewed as a primary method of cancer therapy—it can be used alone or in conjunction with chemotherapy to achieve promising chemo-photothermal therapy protocols. This review aims to give a background into HA and why it is a successful cancer-targeting component of current CNM-based drug delivery systems.

## 1. Introduction

### 1.1. Hyaluronic Acid

Hyaluronic acid (HA) is a biocompatible, nonimmunogenic, and biodegradable anionic natural polysaccharide [1]. It consists of d-glucuronic acid and *N*-acetyl-d-glucosamine connected by alternating by β-(1→4) & β-(1→3) glycosidic linkages [2] (Figure 1).

HA is naturally present in the human body and is a critical component of the extracellular matrix and body fluid, functioning as a regulator for normal structural integrity and development along with regulating tissue in response to injury, repair, and regeneration [3,4]. Its properties make HA a valuable candidate for biomedical applications. The high zeta potential, quantified in a paper by Cavalcanti et al. [5], confirms its highly hydrophilic properties, enabling HA to simultaneously confer water solubility to the overall hybrid delivery system in addition to its primary function as a targeting agent. In particular, Cavalcanti and co-workers showed how the presence of HA led to an increased zeta potential and thus water dispersibility in the HA/soy peptone synthetic mixture, showing its potential for hybrid drug delivery systems [5]. In addition, HA is susceptible to pH-induced structural changes. This finding concurs with studies into the use of HA under different pH conditions, where cumulative anticancer drug release was investigated by circular dichroism experiments. It was discovered that, at a pH of 3, structures such as double helices appeared, while random coils occurred under physiological conditions (pH 7) due to HA being a highly anionic polyelectrode [5]. HA is commonly used for the targeting of tumour cells because of its affinity to target particular over-expressed cell receptors—mainly to the cell surface receptors called hyaladherins, such as clusters of differentiation, commonly cluster of differentiation 44 (CD44^+^) as well as 36 (CD36), protein phosphatase 2 (PP2A), cyclin dependant kinase 9 (CDK9) [6], a receptor for hyaluronate-mediated mortality (RHAMM) [7], lymphatic vessel endothelial HA receptor (LYVE-1) [8], and tumour necrosis factor-stimulated gene-6 (TSG-6) [9,10].

Since CD44^+^ receptors are commonly overexpressed in tumour cells, HA becomes a necessary and powerful targeting component in a drug delivery nanocomposite. CD44^+^ receptors bind to hyaluronic acid and act as an adhesion regulator [11], where it operates in haematopoiesis and lymphocyte activation [12]. Since the purpose of HA is to act as a targeting agent, the practical targeting ability can be compared from one nanocomposite to the other. The most reliable approach for this would be the relative tumour volume reduction over a designated period. In this review, the comparison approach is used to extrapolate approximate results on the relative effectiveness of different nanocomposites bearing HA as a targeting agent and loaded with an anticancer drug for chemotherapy or with a photosensitizer for photothermal therapy.

Figure 2 provides a detailed but simplified visual representation of the HA receptor-mediated endocytosis [13], where HA-decorated graphene oxide nanosheets (HSG) are loaded with Doxorubicin (DOX). The HA is selectively targeted by tumour cell receptors such as CD44. The process proceeds with the accumulation of HSG-DOX within the tumour site followed by the receptor-mediated cellular internalisation. Then, hyaluronidase (HAAse)-mediated HA degradation breaks apart the endosome within the cell and a NIR irradiation allows for an endo/lysosomal escape, eventually leading to the tumour inhibition. Cytotoxicity by this method is therefore directed toward the nucleus of the tumour cell as opposed to directing at non-malignant cells. This process of receptor-mediated endocytosis shows the necessity of the HA component in the hybrid drug delivery system.

HA has surged in popularity within biological fields over the past two decades as the potential role of HA for the development of novel therapeutic strategies for many diseases has been discovered [14]. HA has become well known for many different biomedical applications and treatments. Most commonly, HA is known for its ability to moisturise skin and prevent skin aging [14]. Most of the body’s hyaluronan is found within the skin, with the synthesis of HA increasing during tissue injury and wound healing [15]. It has been shown that HA in conjugation with drugs has excellent efficacy in vivo. The first reports of HA–drug conjugates were published by Akima et al. [16] in 1996. Their study details how HA can be used to enhance the delivery of antitumour drugs into regional lymph nodes and cancerous tissue via a hyaluronate receptor after intravenous (iv), intra-articular (ia), subcutaneous (sc), or intramuscular (im) administrations. Since then, the possibility of active targeting has been investigated by numerous researchers using HA due to its affinity for overexpressed tumorous cells. Biodegradability, biocompatibility, low toxicity, and selective targeting to focus sites enable HA to possess great potential for biomedical and pharmaceutical applications [13,17]. As this is a relatively new concept, most of the literature surrounding conjugated drug delivery has been published in the last ten years, with different drug delivery systems being developed.

Doxorubicin (DOX), one of the most used anticancer drugs, is a highly researched drug molecule that was first identified for its anticancer abilities in 1950 and was clinically approved in 1963, having been proven effective in vivo as an anticancer agent [18]. DOX doesn’t block existing cancerous growth but instead works by blocking an enzyme called topo isomerase 2, which is needed by cancer to divide and grow. To boost anticancer activity, DOX is often used in conjunction with other cancer drugs, such as the combination with Gamitrinib [19]. Gemcitabine (GEM) has also been recorded many times within the literature as a valuable anticancer drug [20]. GEM is a nucleoside metabolic inhibitor that works by slowing down the growth of cancer cells, which then kills them [21], and it has shown to be quite effective in treating several types of cancers such as cell lung cancers, pancreatic cancers, breast cancer, ovarian cancer, bladder cancer, head and neck cancer, cervical cancer, and renal cancer [20]. Several other types of anticancer drugs were used within the literature, such as epirubicin [22], mitoxantrone [23], quercetin [24,25], camptothecin [26], carboplatin [27], irinotecan [28], metformin [29], chlorin e6 [30], SNX-2112 [31], and salinomycin [32].

Near-infrared laser (NIR) photoablation is an important tool in the area of cancer therapy, as the NIR radiation efficiently penetrates throughout the tissues without harmful effects on healthy cells [33]. NIR lasers are typically used in conjunction with nanocomposites to actively target the tumorous cells, producing their selective death by photoablation, as schematically shown in Figure 3.

In a typical photoablation experiment, a NIR laser is commonly focused on a particular point of interest, containing a high concentration of photosensitisers [33]. A photosensitiser is any non-toxic molecule that can be activated by light and generates molecular oxygen that can damage cellular structures [34,35]. The laser induces localised heating that eventually leads to cell death. The use of this technique allows for the elimination of tumorous cells by exploiting the affinity of HA to the HA receptor located on such tumorous cells. The HA, in conjunction with a specific carrier, can then be efficiently targeted by the laser, removing both HA and HA receptors. Photothermal therapy is a highly effective cancer treatment method that uses a photosensitizer to irradicate tumorous cells through targeted ablation, as shown in Figure 3 [33].

Photothermal therapy can be used alone as an efficient method of removing tumour volume and can also be used in conjunction with an anticancer drug in a process known as chemo-photothermal ablation. While photothermal ablation has been proven to be efficient on its own [33], there are limitations to the cancers that can be treated by this method due to the interference of skull thickness or adipose tissue thickness (ATT) on brain or muscle treatments [36]. Overall, the photothermal ablation method is a highly regarded method for tumour volume reduction and, used in conjunction with the targeting ability of HA, becomes one of the most effective techniques for removing tumour growth.

### 1.2. Carbon Nanomaterials and Interactions with Hyaluronic Acid

Carbon nanomaterials (CNMs)—a subclass of nanomaterials—have been utilised in several different fields, including biomedicine [37]. Within biomedicine, they have a number of specific applications, including acting as a carrier for traditional drugs to prevent the rise of drug resistance [37,38]. Drug resistance has always been a problem but has been reported to be mounting [38].

Several CNM-based drug carriers have been investigated in conjunction with HA, including carbon nanotubes (CNTs), graphene, graphene oxide (GO), graphene quantum dots (GQDs), and carbon nano-onions (CNOs) (Figure 4). In this review, we cover CNMs classes with multiple literature-case examples where HA was utilised to impart targeting ability onto the system. Other CNMs such as CNOs, which have only one literature case where they were functionalised with HA [39], have been excluded from this review.

Aggregation in aqueous solutions, driven by intermolecular interactions, has been a challenge for many CNMs due to the hydrophobic nature of their pristine carbon surface. As with any carbon structure, CNMs are highly hydrophobic and have to be made hydrophilic for biological applications. While this presents its challenges, the benefits of utilising these biocompatible CNMs for drug delivery systems in vivo outweigh the key fundamental issues related to their hydrophobic nature.

In recent years, a number of efficient surface modification approaches has been developed to enhance the solubility of CNMs. Covalent strategies have the advantage to introduce onto the CNM surface polar groups such as carboxylic acid groups, leading to a water soluble and stable derivatives [41]. However, this strategy leads to the disruption of the regular sp^2^-hybridized network, thus potentially affecting the intrinsic properties of the materials [42,43]. Therefore, in order to preserve the pristine surface of CNMs and thus their native physicochemical properties, non-covalent approaches are typically preferred [44,45].

A standard non-covalent method for improving the dispersibility of CNMs is through π–π stacking [46]. This non-covalent approach is favoured over covalent interactions as it enables the CNMs to maintain their original pristine properties while promoting their dispersion in water. It has been proposed that, since both gemcitabine and single-walled carbon nanotubes (SWCNT) contain conjugated aromatic rings, π–π stacking interactions are expected to be established between the cystine ring of the gemcitabine and the SWCNT inner wall surface [47].

Options other than π–π stacking are also available, but the non-covalent functionalization of the CNMs surface is preferrable. The non-covalent attachment of HA to CNMs would favour stronger interactions over π–π stacking. Phospholipid structures have been used to preserve the pristine CNM carbon surface and enable non-covalent attachment. Several papers used this method, where structures such as 1,2-dimyristoyl-*sn*-glycerol-3-phosphoethanolamine (DMPE) [39], polyethyleneimine (PEI) [48], polyethylene glycol (PEG) [49,50], 2,2′-(ethylenedioxy)bis(ethyleneamine) (EDBE) [7], spiropyran [51], adipic acid dihydrazine (ADH) [52,53], poly(maleic anhydride-alt-1-octadecene) (PMAO) [54], carboxymethyl chitosan (CMC) [55], polyethylene oxide (PEO) [56], along with several other synthetic structures, proved to be relevant for the functionalisation of CNMs with HA.

HA can be linked to drugs or drug carriers and can improve retention times and the half-life of a drug, as was seen for Insulin by Chu et al. [57,58]. Effective interactions between HA and various CNMs have been hypothesised, and phospholipid structures have been used to overcome the highly hydrophilic and hydrophobic nature of HA and CNMs, respectively. Regarding graphene, computational calculations predicted that CH–π and OH–π interactions are formed primarily between HA and unmodified graphene [59]. On the other hand, there are a large number of OH–O and NH–O interactions between HA and GO, as HA is a hydrophilic molecule.

For biomedical applications where a moderate interaction strength could be required, tailoring interactions between biomolecules and graphene is the best option [60,61]. Wang et al. [59] detailed various possibilities and strengths of interactions and suggested that graphene functionalised with OH, COOH, O-containing, N-containing, or NO-containing groups would be appropriate for a moderate interaction strength. Such modifications could be implemented to improve the biomedical application of GO-conjugated HA. The use of N-doped GQDs in conjunction with HA is discussed by Campbell et al. [62]. The nanocomposite is covalently bound to ferrocene, which selectively targets cancer cells and causes the generation of reactive oxygen species [63] that are cytotoxic to cells.

## 2. Carbon Nanomaterial Conjugated with Hyaluronic Acid for Drug Delivery

In this section, we compare and critically discuss the results published in the literature focusing on how effective the overall nanocomposite, consisting of HA and a specific CNM, was at removing tumours. Some results did not show a clear tumour reduction profile. In this case, the cumulative anticancer drug release profiles and cell viability were compared. While this would not be as effective an approach, the cell viability indicated how effective the HA was. Different processes such as photoablation and photothermal therapy, along with the chemotherapy, are also discussed and the results compared. For our purposes, we decided to divide this section depending on the CNM discussed. HA, being the targeting agent, would be a critical component for the relative tumour volume reduction profiles, as it is an indication of the potential for receptor-mediated endocytosis. Therefore, this metric is the most commonly discussed. Cellular viability would be altered by the presence of HA and therefore this metric must also be examined. It must first be established that the HA hybrid delivery system is biocompatible with the relevant non-malignant cells, and then tumorous cells can be examined to determine if the HA has impeded or improved the overall cellular viability. Finally, fluorescent emission is another beneficial metric, as it clearly details where the hybrid system, directed by the HA, is likely to accumulate.

### 2.1. Carbon Nanotubes

Carbon nanotubes (CNTs) are large cylindrical molecules consisting of a hexagonal arrangement of sp^2^-hybridised carbon atoms, which may be formed by rolling a single sheet of graphene, called single-walled carbon nanotubes (SWCNTs), or by rolling up multiple sheets of graphene, named multi-walled carbon nanotubes (MWCNTs) [64]. While these might be two completely different categories with respect to the chemistry of the CNTs, the experimental results did not appear to differ significantly in the relative tumour volume reduction or the cumulative drug release profiles of the respective anticancer drugs and therefore SWCNTs and MWCNTs are discussed under the same headings, with similar nanocomposites prepared for both. Various factors contribute to an effective drug delivery system using HA as a targeting agent. Primarily, the anticancer drug is the main component that affects tumuor removal since this has the chemotherapy effect and is the active component. The effectiveness of the HA targeting ability is also important. Different M_w_ (6.4, 17, 51, 200 and 1500 kDa) HA were investigated by Arpicco et al. [65], using a non-covalently bound phospholipid structure 1,2-dimyristoyl-sn-glycero-3-phosphoethanolamine (DMPE) to HA and SWCNTs, loaded with DOX (DOX/CNT/HA-DMPE). The hybrid nanocomposite with a HA M_w_ of 200 kDa resulted in a better targeting ability and drug release profiles than any of the other systems. In particular, the material showed a cumulative DOX release of ~7% at pH 7.4 and ~18% at pH 5.5 for 200 kDa, while the other HA M_w_ examined showed a DOX release of only ~4% at pH 7.4 and ~5 to 10% at pH 5.5 [65]. There are several targets for HA receptors similar to those found in tumorous cells, typically CD44^+^, CD36, PP2A, CDK9 [6], RHAMM [7], LYVE-1 [8], and TSG-6 [9,10]. Specificity for these overexpressed receptors on tumorous cells is imperative to prevent cytotoxic effects in other biological systems. Utilising a hybrid system consisting of chitosan/rhodamine B-hyaluronic acid-paclitaxel nanoparticles (CS/RB-HA-PTX NPs), Li et al. [66] were able to determine the specificity of the hybrid system targeting ability, with Rhodamine B (RB) being utilized for its intrinisic fluorescence spectrum. RB was also used alone as a control to demonstrate the background level of fluorescent intensity without the tumour targeting ability. The effectiveness of the targeted delivery of HA can be seen in Figure 5, showing the imaging of tumour-bearing mice and their respective organs and tumour when comparing an anticancer drug alone to the nanocomposite CS/RB-HA-PTX NPs. The scale along the y axis indicates the fluorescent intensity of the hybrid labelled with RB. From Figure 5, it can be seen that the highest fluorescent intensities are in fact seen within the tumour, implying that the specificity of the targeting agent, HA, is effective at delivering the hybrid system to the correct location due to the overexpressed receptor targets. However, specificity could be improved, as the intensities for the stomach and the intestine are both high after oral administration. In comparison to the RB alone, the liver, spleen, kidney, and heart all have greatly reduced intensities, while the tumour is vastly increased. From this perspective, the specificity is highly improved upon when compared to a non-targeted approach.

Tumour volume reduction is imperative to the success of HA-conjugated CNMs and their potential as hybrid drug delivery systems. In particular, Bhirde et al. [67] showed an exceptional use of chemo-photothermal therapy using a cholinic acid-derivatized hyaluronic acid and semiconducting single walled carbon nanotube loaded with Doxorubicin (CAHA-sSWCNT-DOX). The CAHA component of the hybrid provided extra stability in vivo and could undergo versatile chemical modification and eradicate the tumour growth after only two days. The cell viability of the ovarian cancer cell line (OVCAR8) and the ovarian cancer cell line/Adriamycin resistant cell line (OVCAR8/ADR) cells dropped to almost 0% after photothermal therapy, making this paper highly effective at treating tumour growth. On the other side, in the absence of a NIR irradiation, the hybrid showed a cell viability of <10% and 60% in OVCAR8 and OVCAR8/ADR, respectively. The strong optical absorption of CNTs in the near-infrared biological widow and their drug delivery abilities enabled the eradication of multi-drug resistance tumors in vivo with a single dose of drug in combination with PTT.

There are a number of factors that make this hybrid so effective. First, the CAHA-sSWCNTs composite is quite stable without any aggregation over time; second, the ssWNCTs are able to act not only as the drug carrier but their strong optical absorption can also be utilized for the PTT; finally, the combination of DOX in conjunction with PTT is an effective combination, as it prevents not only the growth of the tumour but also eradicates the existing tumour cells [67].

Several combinations of nanocomposites and anticancer drugs can be investigated to augment the observed anticancer effects [19]. In a paper reported by Yao et al. [68], Epirubicin was selected as the drug in conjunction with a carrier consisting in SWCNTs functionalized with disteraroylphosphatidylethanolamine-hyaluronic acid (EPI-SWCNTs-DSPE-HA). In another work reported by Datir et al. [7], the multi-walled counterparts was utilized and functionalized with a HA-2,2′-(ethylene dioxy)bis(ethylene amine) derivative and then loaded with Doxorubicin (HA-EDBE-MWCNT-DOX). In both cases, the engineered nanocomposites were successful to target the cancer cells overexpressing the HA receptors. In particular, the HA-EDBE-MWCNT-DOX hybrid showed no evidence of increased tumour volume after 6 days post-injection and was comparable to the control group. After 10 days of treatment, it was evident that cardiotoxicity levels were increased significantly in animal groups treated with free DOX. Conversely, the cardiotoxicity of the CNT-treated groups showed insignificant differences with the control group, essentially eliminating the risk DOX poses to cardiotoxicity on biological systems [7]. Overall, this is an effective combination to enhance the tumour targeting ability and reduce the cardiotoxicity seen from the anticancer drug DOX alone.

In another paper reported by Liu et al. [69], the authors compared the effectiveness at reducing the human breast tumour volume (MDA-MB-231) of free DOX to that of a system composed of DOX-loaded SWCNTs functionalized with Hyaluronic acid (SWCNT-DOX-HA). After five days, the MDA-MB-231 spherical tumour volume was greatly reduced for the free DOX, SWCNTs-DOX, and the SWCNTs-DOX-HA, meaning that even though the nanocomposite could be used for a more targeted approach, it was almost as effective when applied directly to the tumour. The migration index, which indicates how cells move to change and reach their proper position to execute their function [70], showed that free DOX was less than half that of SWCNT-DOX-HA and the apoptosis rate was approximately half for SWCNT-DOX-HA (<40%) as it was for DOX (~75%); however, this is far superior to the SWCNT-DOX nanocomposite (~10%) with a background control value of 2%. The in vitro tumorous spheroids became irregular and smaller, indicating that the SWCNTs-DOX-HA could penetrate deep into the centre of the cell to induce cell aptosis [69].

#### 2.1.1. Cumulative Release Profiles of HA-Conjugated CNTs Loaded with Anticancer Drugs

Several papers used varying methods for the cumulative release profiles of the anticancer drug, with some being effective at all pH conditions, some being pH specific, and some having a general mix for both fast and slow-release profiles.

An example of a pH-triggered drug release profile is seen in a paper by Mo et al. [8], where single walled carbon nanotubes were functionalized with chitosan and a 6 kDa HA and then loaded with Doxorubicin (SWNTs-CHI-HA-DOX). The nanocomposite showed to be stable at a pH of 7.4, with virtually no drug release seen after 72 h (Figure 6A). However, this nanocomposite is exceptionally efficient when introduced to conditions imitating that of lysosomal conditions (i.e., at pH 5.5), where a cumulative release of 85% was seen after the same period (Figure 6B). It is accepted that there is a lower release profile when considering conditions at a physiological pH of 7.4 compared to a more acidic pH such as a pH of 5.5. Chitosan is sensitive to pH, and this could be the influencing factor as to why there is such a noticeable increase in its effectiveness. Similarly, to keep drug release profiles high at pH conditions of 7.4, one paper utilised α-tocopheryl succinate and >1000 kDa HA functionalized MWCNTs loaded with DOX (α-TOS-HA-MWCNTs/DOX) to improve the cumulative release of DOX at all pH conditions [71]. The release across all nanocomposites was quite high, as it showed a steady increase in cumulative release, nearly reaching 20% in comparison to the 5% release seen for a very similar formulation in the paper by Mo et al. [8]. Variations in the cumulative drug release profiles were seen in several papers. Based on the SWNT-CHI-HA-DOX release profile seen in Figure 6, it would be expected that single-walled nanotubes loaded with chitosan-hyaluronic acid and paclitaxel (SWNT-CHI-HA-PTX) would have a similar release profile.

However, a paper by Yu et al. [72] used single walled carbon nanotubes functionalized with chitosan and hyaluronic acid and loaded with Paclitaxel (SWNTs-CHI-HA-PTX) and they produced a much shorter release profile for the majority of the cumulative release of PTX. The quick cumulative release profile, releasing 60% out of 70% within the first 2–3 h, could be attributed to the 6 kDa HA that has been shown to have slower release profiles for higher M_w_. However, since chitosan has been previously used for pH-sensitive drug release, it would not have been expected that the pH 7.4 would release 40% of PTX. This result contradicts the pH-dependance of chitosan as seen in the paper by Mo et al. [8], where chitosan has a much more significant effect on implementing pH-dependent drug release, and further investigations could be explored. The correlation of higher M_w_ HA having slow-release profiles also continues the trend between the papers by Mo et al. and Singhai et al. [71], as the profile by Mo et al. extended over >5 h before levelling off at a pH of 7.4 while the study by Singhai et al. [71] extended over 125 h and appeared as if it had not levelled off.

A paper presented by Prajapati et al. [49] using gemcitabine loaded onto HA-multi-walled carbon nanotubes (GEM/HA-MWCNTs) has an exceptional release profile at pH 7.4 (Figure 7), with a cumulative release of >80% seen after the 144-h study.

Figure 6 and Figure 7 are complete reversals of each other from a pH controlled drug release perspective, as, in Figure 6, the drug release is completely controlled by the pH and the cumulative drug release is only really seen at a pH 5.5, with an 18 fold increase from the 5% release at a pH of 7.4. Figure 7 is not highly specific to the pH for either measurements, with >5% change between the cumulative release seen at a pH of 5.3 and the release seen at a pH of 7.4. Under certain circumstances, it is preferable if the nanocomposite would not be affected by pH, as this could be more inclusive for various cancer treatments. However, pH targeted drug delivery can be preferential under specific conditions; the possibility of cumulative release being higher for pH 5.3 or pH 7.4, dependent on constituents of the hybrid drug delivery system, leaves universal possibilities for nanomaterial hybrid chemotherapy treatment rather than only pH specific treatments.

#### 2.1.2. Relative Tumour Volume Reduction

The binding of HA to the CNMs is essential to effectively carry the anticancer drugs to the tumour sites. A few papers showed a lower relative tumour reduction than expected, and this could be due to the inefficient binding of the HA to the CNM surface. A study by Hou et al. [73] compared the effects of the functionalization of a 46 kDa HA to different CNMs, namely, SWNTs, GO, and a C_60_ fullerene for photothermal ablation applications. While the control sample showed a relative tumour volume increase of 700% after 10 days, the nanohybrid formulations were all virtually ineffective, with the greatest inhibition being the hyaluronic acid-conjugated single walled carbon nanotube in conjunction with a 808 nm laser. In particular, it showed a 450% increase in tumour volume after the same time period. The antitumor effect of all the nanohybrid systems for photothermal therapy was essentially inexistent, suggesting there could have been a common issue across all nanohybrid systems. The results of this paper provide evidence of the hypothesis that HA may have not coordinated properly with the CNMs and that the hybrid structures cannot be used efficiently as a targeting system for drug or photosensitizer delivery.

Photothermal ablation using a laser is a faster and effective technique at eradicating the tumour. When photosensitizers are present in significant quantity around the overexpressed HA-receptor tumour cells, the NIR radiation can be applied for a full ablation effect. This has the ability to induce tumour inhibition not only to a level where tumour growth would be prevented but to a level where the tumour would also be reduced in size. Phototherapy is a proven technique in many different papers, with efficient relative tumour reduction [28,74]. However, the paper by Hou et al. [73] showed virtually no noticeable difference in tumour volume when using photothermal ablation therapy compared to the blank control. The most likely issue is the covalent bonding, which would be predicted to occur between HA and the CNM. The attachment of HA to the pristine surface of these CNMs (SWNT, GO, and C_60_) involved covalent attachment rather than the preferable non-covalent approach. The use of a phospholipid structure to act as a bridging material between the highly hydrophobic CNM surface and the hydrophilic HA would be beneficial from this point of view, with π–π stacking being available for surface attachment without modifying the intrinsic properties of the CNMs.

Two possibilities could be used in conjunction with a phospholipid to improve the efficiency of the nanocomposites. The first method could be to use more than one wavelength for the laser [30]. The second is to use a combined chemo-photothermal therapy approach that has proven to be quite effective in several papers at removing tumour growth, and the preposition of DOX-loaded CNMs could be quite effective, as DOX is known for preventing the further growth of cancerous cells by blocking topo isomerase 2, preventing cancerous cells from dividing and growing [18].

The current challenge in nanomedicine is defining which combination of CNMs and anticancer drugs is most effective, as there are so many effective combinations seen in this review. An interesting example is a paper by Yao et al. [32], where Salinomycin-loaded Single walled nanotubes functionalized with chitosan and hyaluronic acid (SAL-SWNTs-CHI-HA) was utilized. Notably, this paper used Salinomycin (SAL), not as commonly seen in conjunction with the other CNM nanocomposites within the literature, showing the opportunities that SAL offers as an anticancer drug. The results showed it to be quite effective in reducing tumour volume. From a chemotherapy point of view, when the possibility of treating tumours for photothermal ablation isn’t available, the composite described by Yao et al. [32] is exceptionally effective, reaching a 18.2 ± 1.2% relative tumour volume after day 6 for the SAL-SWNTs-CHI-HA, while the control reached 433.3 ± 6% in the same time period. There is a strong possibility that the mammospheres (mammary epithelial stem cell aggregates, derived from breast tumours) were penetrated to the centre by the drug delivery system because of their granular and irregular shape on the outside and finally broke into pieces, similar to what was seen by Liu et al. [69].

In the area of Carbon Nano Onions (CNOs), there have only been two papers directly relating to hyaluronic acid [39,75]. While these papers show good promise for HA-conjugated CNO as a potential drug carrier for targeted drug delivery, the lack of literature proved challenging to make any direct comparisons. Concerning the results, which could be compared to other CNMs, the cell viability is relatively high, and the confocal imaging shows that the HA is quite effective at localising around the receptors. Both papers demonstrate the ability of the CNOs to disperse in aqueous media. The paper by Zhang et al. [75] even stated that all the nanomaterial dispersions were stable, with no evidence of precipitation over several weeks. Prospects would be high in the area of CNOs in conjunction with HA based on previous biocompatibility testing. However, the lack of data surrounding CNOs’ relationship to HA leaves such bio-applications inconclusive.

Opposed to the results seen from photothermal therapy, which will be seen in the following section for graphene, a paper by Shi et al. [76] utilizing hematoporphyrin monomethyl ether and hyaluronic acid functionalized carbon nanotubes with 532 and 808 nm lasers (HMME-HA-CNTs) noted no reduction in tumour volume after 9 days. This result comes as a surprise, as HMME is a promising photosensitizer [77]. Similar results are also seen in a paper by Hou et al. [78], where Gadolinium doped Single walled carbon nanotubes functionalized with HA (M_w_ of 12 kDa) linked by a disulfide bond to Doxorubicin (GD/SWCNTs-HA-ss-DOX) is used for a combined chemo-phototherapy. In general, chemo-photothermal papers are regarded as the most efficient at reducing tumour volume either entirely or nearly entirely. However, the tumour volume either remained constant or slightly below the original tumour volume on the 9^th^ day in this specific case. Primarily, the main question that would be asked would be surrounding the targeting ability of the HA. In vivo imaging shows high fluorescence surrounding the tumour but also high fluorescence in other areas showing high but not total specificity, possibly causing a lack of hybrids surrounding the tumour. Since there was no tumour growth, it is probable that the anticancer drug DOX was successful in inhibiting the tumorous cells’ further division and growth.

### 2.2. Graphene, Graphene Oxide and Graphene Quantum Dots

Graphene, graphene oxide (GO) and graphene quantum dots (GQDs) are valuable members of the carbon family. Graphene is formed by a thick sheet of carbon atoms bonded by sp^2^ hybridisation arranged in a hexagonal array [79]. Graphene oxide (GO) is a unique material that can be viewed as a single monomolecular layer of graphite with various oxygen-containing functionalities such as epoxide, carbonyl carboxyl, and hydroxyl groups [80]. Graphene quantum dots (GQDs) are composed of a few layers of graphene fragments, typically less than 10 nm in size [81]. They are zero-dimensional members of the carbon family and are usually considered a chopped graphene sheet fragment [82]. Graphene and its derivatives have attracted considerable interest due to their unique 2-D structure that provides a large surface area for π–π stacking [83,84,85,86]. The functional groups attached to GO sheets such as epoxy, hydroxyl, and carboxylic acid facilitate easy modification [87] and facilitate the dispersion of GO in specific polymer matrices or polar monomers during in situ polymerization [88].

Figure 8 shows the fluorescent intensity of a control sample, Graphene Quantum Dots (GQDs), and HA-conjugated Graphene Quantum Dots (GQD-HA). The intensity corresponds to the volume of the drug carrier and therefore the drug reaching the target site. From the comparison of the intensity, it can be seen that the GQD-HA localizes much more in the target site than GQD alone, suggesting that the presence of HA is required to maximize the delivery of drugs inside the tumourous cells [89].

Similarly to CNTs, a comparison is made in terms of the relative tumour volume reduction for the graphene family materials. In the cases where the relative tumour reduction profiles were missing, the cumulative anticancer drug profiles and the relative cell viability were analysed to interpret how effective the HA nanocomposites would be at the target delivery to either the overexpressed CD44^+^ receptors or other tumorous sites. In general, the relative tumour volume reduction profiles are quite a reliable method for the comparison of the hybrid drug delivery systems, as they not only quantify the effectiveness of the anticancer drug but also account for the targeting ability of the HA and the joined coordination of the overall hybrid system.

#### 2.2.1. Chemotherapy Using HA-Conjugated Graphene, Graphene Oxide, and Graphene Quantum Dots Delivery Systems

Chemotherapy focuses purely on the anticancer drug delivery to the tumour target site using HA as the targeting agent and a CNM as the carrier. In the literature, there is a broad range of HA M_w_ being reported. Yi Teng Fong et al. [90] and Kaya et al. [56] reported studies with HA M_w_ values of 1800 and 800 kDa used, respectively, while, in a paper by Basu et al. [29], a HA M_w_ of 8–15 kDa is employed. The HA Mw may have significant effects on the hybrid’s stability and potential as a targeting agent.

In the paper by Basu et al. [29], two nanohybrids are compared in terms of cell viability and cell death. These nanohybrids consist of NH_2_-polyethylene glycol-poly(d,l-lactide-co-glycolide) GO loaded with metformin (NH_2_-PEG-PLGA-GO-MET), and hyaluronic acid functionalized GO loaded with metformin (HA-GO-MET). The cell death of the epithelial-like breast cancer cells (MDA-MB-231) is compared to the percentage of survival of the peripheral blood mononuclear cells (PBMC) and the non-malignant breast epithelial cells (MCF10A) for the two nanohybrids. Interestingly, the cell death of MDA-MB-231 cells from HA-GO-MET is approximately double that of NH_2_-PEG-PLGA-GO-MET at nearly all concentrations up to a maximum concentration tested of 45 μg/mL In particular, HA-GO-MET showed a cell death of MDA-MB-231 of 90% (cell viability of 10%) and a cell viability of both PBMC and MCF10A of >80%. From these results, it would have been expected that the overall relative tumour volume reduction would have been much higher, or, at a minimum, it would not have increased. Possibly, the addition of HA to GO, prepared by the Hummers method [91], was not a stable combination for in vivo analysis, and the addition of one of the phospholipid structures found in several other papers in the literature in conjunction with the targeting ability of HA could have been implemented, since the cell viability of MET was so successful. As seen in Figure 9, an in vivo experiment by Yang et al. [92] utilizing Doxorubicin-loaded paramagnetic QDs functionalized with hyaluronic acid (DOX/PGQD-HA) to determine the time taken for the nanohybrid to effectively target the human lung cancer cell A549 tumour site, where the nanohybrid could be observed using magnetic resonance imaging (MRI). After 30 min, the presence of the nanocomposite was visible in the relevant area, and, after 2 h, it was concluded that a significant volume of the PGQD-HA with DOX removed for the imaging process had reached the target site. The imaging half an hour and 2 h post-injection is a valuable marker for how long it would take for the effects of the nanohybrid to be seen post-injection. Figure 9 also shows the gradual process of the human lung cancer cell appearing lighter with the increasing post-injection time, indicating the permeation process of the probe in the tumour area. In a paper by Zhang et al. [24], GO was functionalized with polyetheramine and hyaluronic acid (M_w_ of 110 kDa) and loaded with Quercetin (GO-PEA-HA/QUE) as the drug. The Quercetin (Que) anticancer drug showed a high cumulative release but did not present a very effective relative tumour volume reduction compared with many other anticancer hybrids. While the pH was not explicitly mentioned for the cumulative Que release, it was referenced to be the same as a study by Song et al. [53], which utilised HA-GO loaded with doxorubicin (HA-GO-DOX) and had pH values of 5.3 and 7.4.

The cumulative release did not show a significant reduction compared to other values in the literature, but the release rate was unusually fast, with the majority of the release profile occurring in the first 3 h. Comparing this to other papers using CNM hybrids for chemotherapy applications and HA targeting, the release times for approximately the same percentage of the anticancer drug were 20 h by Nigam et al. [93] for human serum albumin and hyaluronic acid functionalized graphene quantum dots loaded with gemcitabine (HSA-GQDs-HA/GEM) and 6 h by Luo et al. [50] utilizing hyaluronic acid functionalized quantum graphene with rhodamine B isothiocyanate and loaded with doxorubicin (HA-Q-G-RBITC/DOX). The second release profile of 6 h was also surprisingly short but showed a continued release of DOX at a pH of 5 for up to 25 h. In another paper, nanographene oxide was decorated with HA (M_w_ of 35 kDa) by a liner cystamine dihydrochloride containing sulfide bonds and loaded with Gefitinib (NGO-ss-HA-Gef) [94]. The quantifiable tumour volume reduction figure showed an increase from initial tumour volume, and, after 24 days, the tumour volume had almost doubled in size. This was a poor result, as the tumour volume was not at a minimum maintained in size. In comparison to the control group, this is an effective result, which grew by 7–8 times the original volume, showing that the nanohybrid had some effect but wouldn’t be an acceptable enough reduction to fully inhibit the tumor growth. There are several reasons that could have caused this nanocomposite to under-perform. One probable cause is the use of a cystamine dihydrochloride used to form the disulphide bonding to attach the HA to the NGO. The cell viability was quite efficient for 24–72 h for the higher concentrations tested, so it is possible that the injection volume was too low. Interestingly, the cumulative release profiles at pH 5.5 and pH 7.4 are almost identical for the hybrids, indicating that this hybrid is not pH dependant, as was the case with several hybrids with chitosan as a component.

#### 2.2.2. Photoablation Therapy Using HA-Conjugated Graphene, Graphene Oxide, and Graphene Quantum Dots Delivery Systems

Photodynamic therapy is a light-sensitive cancer therapy that employs the cooperative interaction between light and photosensitiser to promote tumour suppression [95,96]. The cancerous cell is eventually killed under the correct conditions post-irradiation, as the photosensitizer overheats and essentially is self-terminated along with the neighbouring cells overexpressing hyaluronic acid receptors. There are many papers that used photoablation as their primary source of operation to remove tumour growth.

In a paper by Kim et al. [97] a HA M_w_ of 230 kDa was employed in a system comprising polyethylene glycol-g-poly(dimethylaminoethyl methacrylate)-HA–reduced GO (PgP/HA-rGO) in conjunction with a near-infrared laser. From the results of the relative tumour volume reduction profile, which was carried out over the course of 10 days with an exponential decrease in tumour volume, it could be predicted that, since the tumour volume was only 25% of the original volume, if the study had continued as far as day 12–14 for the PgP/HA-rGO hybrid, the tumour volume could have been removed completely. Regardless, evidence is shown in digital photo images after 20 days, compared to the PBS sample, where the tumour site on the rat has been removed and healed completely. The cell viability to normal Madin-Darby Canine Kidney (MDCK) cells remains high at a 1.0 mg/mL concentration (65%) in comparison to either the epithelial breast cancer cells (MDA MB 231) at ~35% and the human lung cancer cells (A 549) at ~35%. For enhanced and faster results, the tumour photoablation therapy is quite often used in conjunction with an anticancer drug for a combined chemo-photothermal ablation therapy. Khatun et al. [98] chose a graphene functionalized with hyaluronic acid (M_w_ of 7000 kDa) and loaded with Doxorbucin nanogel (GHD nanogel) with the use of a laser for both chemotherapy and photothermal ablation. The nanogel was capable of reducing mice tumour growth gradually to 60% of the original volume after 18 days. Characteristically, photothermal ablation is a faster process than chemotherapy, as the cells are instantly eradicated by the laser. Within the literature, this paper had the second highest M_w_ of HA used, only behind a paper by Hou et al. that had a HA M_w_ of 12,000 kDa [99]. Interestingly, the cumulative dox release continued to increase up to 48 h under in vivo conditions to boast 100% of DOX released under pH 5.0 conditions in the presence of a laser. Since these results showed a slight but consistent decrease in tumour volume, it could be a practical option for lung cancer treatment if the HA at that specific M_w_ is of a more affordable price. The price of specific HA M_w_ is dependent on the quantity produced, so, in theory, if any specific M_w_ were to be mass-produced, the costs would also drop dramatically. Currently, the price is unaffordable for any significant application, with HA costing approximately 150 EUR for every 10 mg [100]. It is predicted that this price could become less than half in a study by Ferreira et al. [101].

In a paper by Jiang et al. [30], the authors used reduced GO-polydopamine on mesoporous silica/chlorin e6/HA and near infrared radiation (rGO-pDA @MS/Ce6/HA), showing a cell viability of almost 0% for the human colorectal carcinoma cell line (HCT-116), using lasers at wavelengths of 808 nm and 660 nm. Given the almost complete HCT-116 eradication, the cell viability of NIH-3T3 is quite impressive, being ~80%. While this is a chemo-photothermal ablation paper, the cell viability on the addition of Ce6 is almost unchanged, making the lasers the primary ablation source.

Utilising photothermal ablation is a difficult technique to perfect, as there are so many different variables such as wavelength used, the components of the nanocomposites, the use of additional chemo or hyperthermia methods, and adjusting test conditions. An interesting nanocomposite hybrid from Wang et al. [102] utilised GQD-capped Prussian blue nanocubes (GQD-PBPGs) with HA and polydopamine. The nanocomposite did not reduce the tumour volume but maintained the original tumour volume; this is still a significant improvement from the control group, which showed a 6-fold increase in tumour volume after the 14 day study period. Several factors could help improve the efficiency and increase the prospects for GQD-PBPGs to reduce tumour volume, such as using more than one wavelength for the laser [30], opting for a photothermal ablation method [103], or possibly using the magnetic hyperthermia system (MHS) technique detailed by Deng et al. [104]. Cell viability graphs showed an outstanding difference to C6 cells with and without the use of the laser, as the cell viability dropped from ~80% to ~10% once the laser was turned on, showing how effective the photothermal ablation method is for this hybrid. Since this cell viability is so high, it could be possible that the selectivity for the tumour was not as high as was necessary.

#### 2.2.3. Combined Chemo-Photothermal Therapy

The combined chemo-photothermal thermal therapy is a popular approach using both an anticancer drug and a photosensitizer in conjunction with a laser for photothermal ablation, giving faster and enhanced tumour treatment. It is quite often seen that a NIR laser with a wavelength of 808 nm is directed toward the tumour for the purpose of photoablation, while the drug carrier, along with an anticancer drug, carries out chemotherapy simultaneously. Combined chemo-photothermal ablation proved to be an effective treatment method, as shown by Zhang [74] and Shao [103]. These papers used the hybrids graphene oxide functionalized with adipicdihydrazide and hyaluronic acid, loaded with methotrexate (GO-ADH-HA-MTX), and mesoporous silica coated with polydopamine reduced graphene oxide and modified with HA, loaded with Doxorubicin (pRGO@MS(DOX)-HA), respectively. The paper by Zhang et al. shows blockage of tumour growth steadily throughout the 22.5 day study period, removing almost complete tumour growth by the end, while the paper by Shao et al. appears to have an almost complete tumour removal after 10 days, both using the combined chemo-photoablation approach with different hybrids. When only the laser is used, photothermal therapy without chemotherapy, or the DOX is added without using the NIR laser, chemotherapy without photothermal therapy, there is a significantly reduced effect where the tumour growth rate is slowed but not to any significant amount, while, when both techniques are used, it appears that there is a minimal amount of tumour volume remaining at the end of the studies for both hybrids. The nanocomposite hybrid used by Zhang, GO-ADH-HA-MTX, appears to be almost as efficient at tumour volume reduction with the use of a laser as the paper by Shao et al. [103] using pRGO@MS(DOX)-HA with a laser. Once again, removing either the laser or the anticancer drug dramatically affected the results, indicating that both hybrid nanocomposites were only effective when used for the combined chemo and photothermal treatment. An example of a similar paper using the chemo-photothermal approach and the hybrid hyaluronic acid and hollow mesoporous carbon nanoparticle functionalized graphene quantum dots loaded with doxorubucin and used in conjunction with a laser at 808 nm (HA-HMCN(DOX)@GQDs) was published by Fang et al. [105]. They used an efficient nanocomposite consisting of GQDs as the nanocarrier, DOX as the anticancer drug, and HA as the targeting agent for tumorous cells in conjunction with the 808 nm laser. After a 21-day study, the tumour volume, although not increasing in volume, had not been reduced with the most effective nanocomposites; the in vivo fluorescent imaging would suggest that the targeting ability of HA was quite effective, as, after 12 h, the HeLa cell was easily identifiable. A possible method of improving this would be to vary wavelength and possibly use more than one wavelength for photoablation. While tumour volume reduction profiles are not always possible, comparisons of cell viability within papers have been quite effective at displaying the effect of certain properties. An example of cell viability showing the difference between photoablation and chemo-photoablation is seen in a paper by Xu et al. using the hybrid nanographene oxide functionalized with hyaluronic acid with gold nanorods and loaded with doxorubicin (NGOHA-AuNRs-DOX) [106]. The chemotherapy approach shows a cell viability rate of approximately 60%, while a result of approximately 40% for chemo-photothermal therapy was seen against hepatoma Huh-7 cells at a conc. of 100 μg/mL. While the cumulative release profiles are not particularly efficient in this paper, the expected tumour reduction would be quite high, as a cell viability rate of 40% is exceptionally high for a 5 min treatment. Another successful paper using chemo-photoablation was a study conducted by Hou et al. [107] utilizing the hybrid Mitoxantrone loaded Graphene Oxide functionalized with hyaluronic acid in conjunction with a laser (MIT/HA-GO), where the relative tumour volume was reduced by 85.19% after 6 days. Since most other studies had a longer study period, it can nearly be guaranteed that the tumour would have been eradicated after a few more days. This result would be extremely efficient since such a large tumour volume reduction in such a short space of time is quite rare. Although the use of a lower M_w_ than the average (12 kDa) has been less efficient in some papers than a M_w_ of 100–300 kDa, there are advantages, such as the cost and prevention of HA meshwork forming that can prevent HA passing through certain physiological barriers [7]. If a method such as chemo-photothermal therapy can be implemented effectively, it appears it is the most effective and cost-friendly approach to treatment. Not all chemo-photothermal therapy removed tumour growth entirely, but it can still reduce tumour growth, as seen in a study by Deng et al. [104]; this paper was slightly different in the sense that the hybrid microcapsule with sodium alginate, chitosan, iron (III) oxide on GO, chitosan, and HA hybrid microcapsule (Alg/Chi/Fe_3_O_4_@GO/Chi/Ha (h-MC)) was used with DOX as the anticancer drug and heated by a process called magnetic hyperthermia system (MHS) along with the use of a NIR laser. There was no fluorescent imaging to show the targeting ability of HA in this paper, which could have been one possible reason why there was no further reduction in tumour volume, as it is not clear if the tumour was targeted successfully. Since both the cumulative release profiles and the cell viability profiles look quite good, it is possible that the HA was not able to target the tumour site as efficiently as planned. Another possible reason is the number of processes running concurrently. This paper is one of the few papers in the literature relating the chemo-photothermal ablation use of HA with the MHS process. Since MHS has been shown to increase cell viability [104], the simplification of the other elements of the nanocomposite such as Alg/Chi/Fe_3_O_4_ could have a possible increase on the overall relative tumour volume reduction. The hyperthermia method is also supported in a paper by Chen et al. [108], which details how apoptosis occurs in response to therapeutics such as cytotoxic drugs, radiation, and hyperthermia [109,110]. Since the combined chemo-photothermal ablation method is a successful technique, the addition of the third hyperthermia method could further increase efficiency to other hybrids used in the literature. Without the MHS, the tumour size is maintained, implying that perhaps the other processes of chemo-photothermal ablation are hindered in some way when compared with the results of other papers.

#### 2.2.4. Dual Receptor Targeting

An interesting approach was taken by Guo et al. [111], which used HA in conjunction with Arg-Gly-Asp (RGD) modified graphene oxide peptides (GO-HA-RGD/DOX). This combination developed a dual receptor targeting system for the delivery of DOX. The performance of the dual receptor targeting ability was quantified with the use of fluorescent imaging. A dramatic increase in mean fluorescent intensity was visible in the region surrounding the SKOV-3 cells on the addition of HA, and then again on the addition of RGD. Since this method of dual targeting can be shown to be more efficient, it could be implemented into other nanocomposites. Comparing the cell viability using GO-HA-RGD/DOX with SKOV-3 cells (target cells) and HOSEpiC cells (normal cells) shows an approximately 20% cell viability for the cancerous cells and an approximately 80% cell viability for the normal cells. The GO-HA/DOX (no RGD) nanocomposite has marginally higher cell viability in normal cells, making it less cytotoxic, but could be covered by the error bar deviations and shows a significantly reduced cell viability to SKOV-3 cancerous cells. In a paper by Zhang et al. [26], the cellular viability of MDA-MB-231 cancer cells in the presence of β-Cyclodextrin functionalized GO-hyaluronated adamantane (CPT@GO-CD/HA-ADA) nanocomposites coupled with the anticancer drug Camptothecin (CPT) were compared to the drug alone. While this paper did not boast a 20% cell viability for the nanomaterial nanocomposites to the cancerous cells, it was more effective (51.5%) than CPT (64.5%), the anticancer drug used alone. Cell viability for the normal fibroblast cells was measured to be 82.5%, similar to that of the first paper by Guo et al. [111] compared to this drug’s non-toxic elements (GO-CD/HA-ADA). The similarity in cell viability results to normal cells only shows the effectiveness of the dual receptor targeting ability, since, ordinarily, the results in the paper by Zhang et al. [26] would be that of an efficiently targeted receptor.

## 3. Conclusions

As discussed in this review, many different options can be explored using hyaluronic acid in conjunction with different carbon nanomaterials. Both carbon nanotubes and the graphene family showed significant effectiveness in tumour volume reduction. Other applicative uses of the HA-CNM systems have been discussed—in particular, using a chemo-photothermal approach can lead to an increased tumour treatment speed. Some of the less-frequently utilised methods discussed in this review can improve the overall efficiency of nanocomposites. Primary examples include (a) using more than one wavelength in photothermal ablation, (b) utilising the magnetic hyperthermia system described in conjunction with both chemo-photothermal therapies, (c) dual receptor targeting, and (d) varying different anticancer drugs to determine the most efficient nanocomposite system.

Overall, HA-conjugated CNMs have largely unrealized potential for the future of cancer treatment. The hybrid systems discussed in this review have displayed tumour volume reduction profiles more efficient than using the anticancer drug alone, linked to the receptor-mediated endocytosis of HA, higher biocompatibility to non-malignant cells, as seen from the cellular viaiblty studies, and selectivity for the tumour from the fluorescent imaging data.

There are still a number of aspects that can be researched further for the biomedical application of HA-conjugated CNMs. Important future research should be focused on selecting the appropriate anticancer drug to be loaded to the carrier, finding the most effective phospholipid structure to bind HA and prevent covalent attachment to the carbonaceous surface, developing novel methods of further reducing the aggregation of the CNMs as well as exploiting the combination of the less-frequently used methods discussed above.

## Figures and Tables

**Figure 1 molecules-27-00048-f001:**
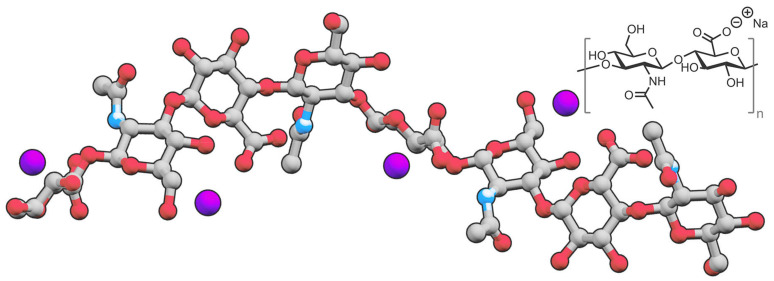
Structure of sodium hyaluronate, a salt derived from hyaluronic acid.

**Figure 2 molecules-27-00048-f002:**
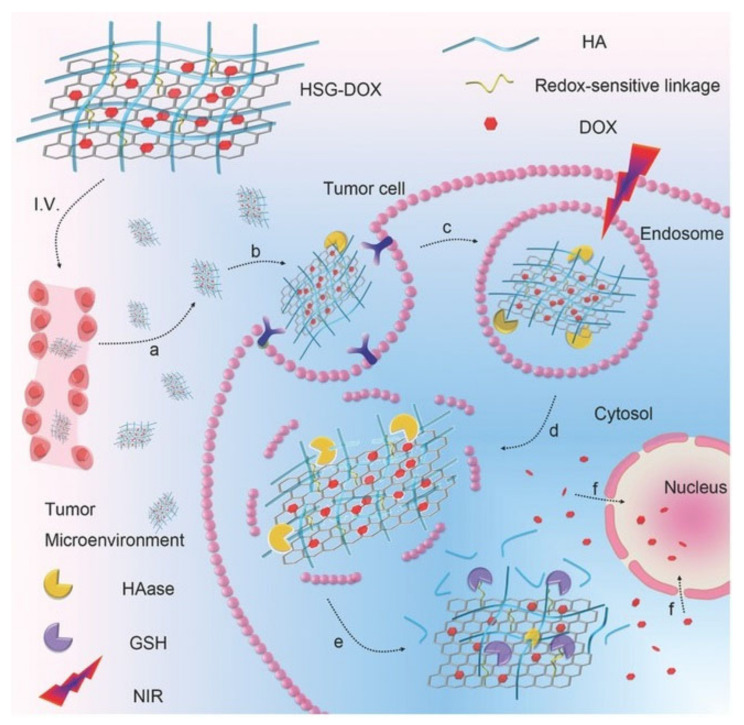
Targeted delivery of the nanocomposite to cancer cell via HA receptor-mediated endocytosis. Adapted with permission from Ref. [13]. Copyright 2017 John Wiley and Sons.

**Figure 3 molecules-27-00048-f003:**
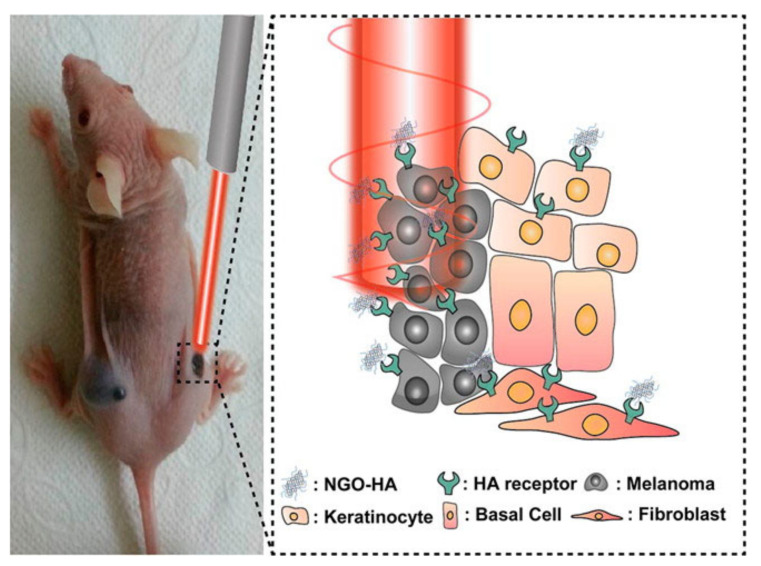
Use of NGO-HA to target HA receptors for photothermal ablation using a near-infrared (NIR) laser. Reprinted with permission from Ref. [33]. Copyright 2014 American Chemical Society.

**Figure 4 molecules-27-00048-f004:**
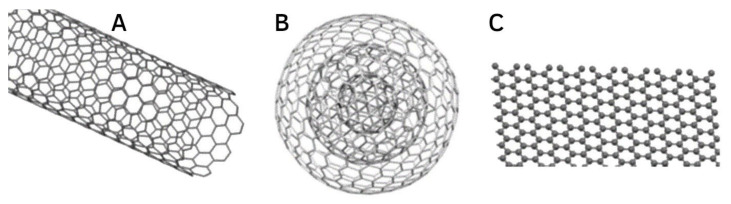
Structures of (**A**) single-walled CNT, (**B**) CNO, and (**C**) graphene. Adapted with permission from Ref. [40]. Copyright 2015 Royal Society of Chemistry.

**Figure 5 molecules-27-00048-f005:**
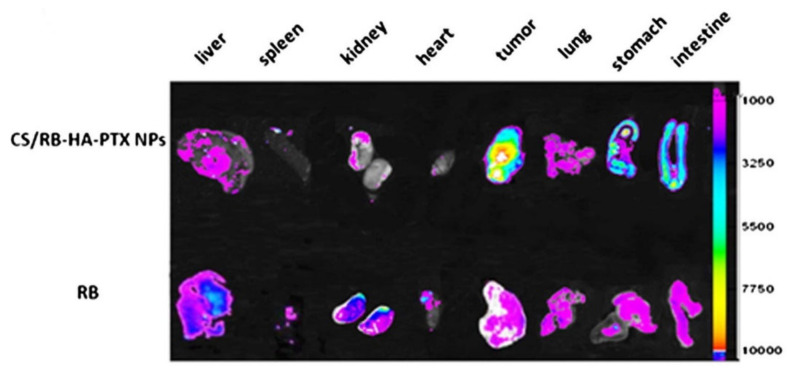
In vivo imaging of the tumour and various organs, showing the specificity of the targeted tumour delivery when using chitosan/rhodamine B-hyaluronic acid-paclitaxel nanoparticles (CS/RB-HA-PTX NPs) over rhodamine B (RB) taken orally. Adapted with permission from Ref. [66]. Copyright 2013 Springer Nature.

**Figure 6 molecules-27-00048-f006:**
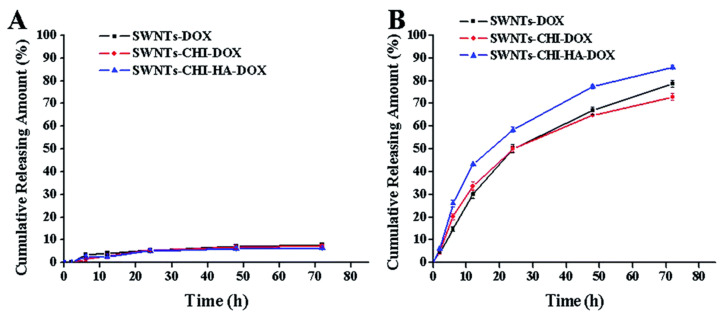
DOX release at 37 °C in (**A**) pH 7.4 and (**B**) pH 5.5 PBS. Reprinted with permission from Ref. [8]. Copyright 2015 Royal Society of Chemistry.

**Figure 7 molecules-27-00048-f007:**
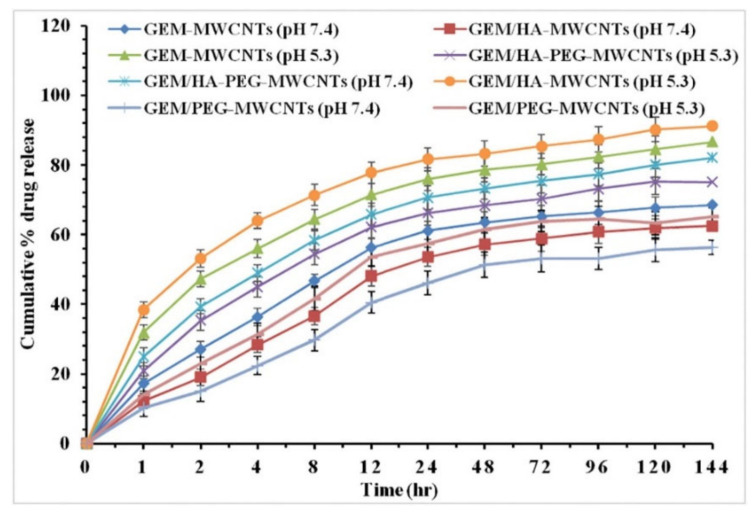
In vitro drug release profiles of GEM-MWCNTs, GEM/HA-MWCNTs, GEM/PEG-MWCNTs, and GEM/HA-PEG-MWCNTs at phosphate buffer (pH 7.4) and lysosomal conditions (pH 5.3). Reprinted with permission from Ref. [49]. Copyright 2019 Elsevier.

**Figure 8 molecules-27-00048-f008:**
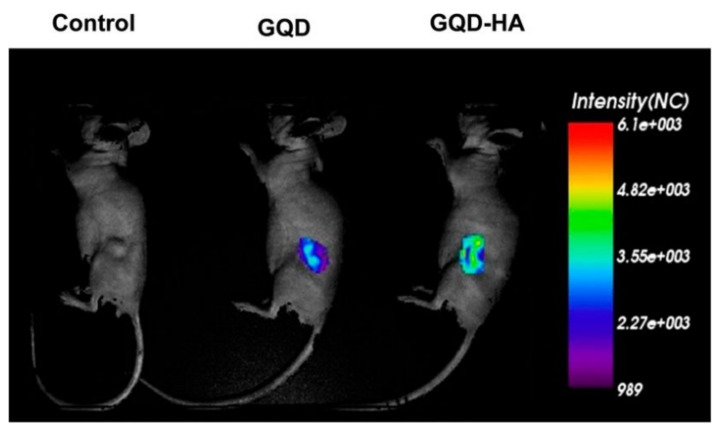
Comparison of HA-targeting ability of GQD-HA to the control and CNM alone. Reprinted with permission from Ref. [89]. Copyright 2013 American Chemical Society.

**Figure 9 molecules-27-00048-f009:**
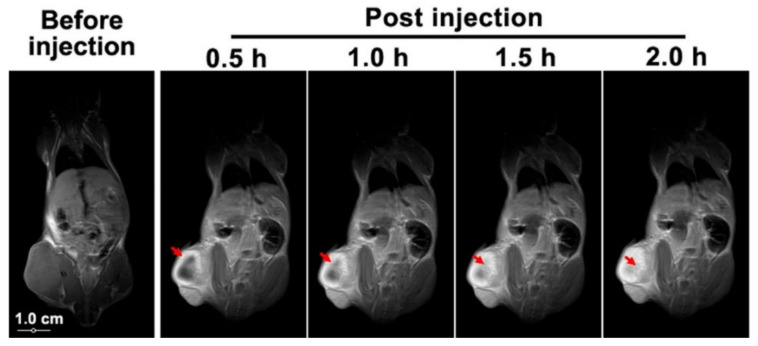
Magnetic resonance of A549 tumour-bearing mice prior and post injection of poly-graphene quantum dots-hyaluronic acid (PGQD-HA). Adapted with permission from Ref. [92]. Copyright 2019 American Chemical Society.

## Data Availability

Not applicable.

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
