# Peer review of "Hyaluronic Acid-Conjugated Carbon Nanomaterials for Enhanced Tumour Targeting Ability"

_molecules, 2021, doi:10.3390/molecules27010048_

Round 1

Reviewer 1 Report

The manuscript entitled “ Hyaluronic Acid-Conjugated Carbon Nanomaterials for Enhanced Tumour Targeting Ability” summarize the latest research on using hyaluronic acid as targeting agent in HA-carbonaceous nanomaterials for tumor chemo and photothermal therapy. The material is very interesting and provide a valuable view on HA ability to act as targeting agent, however there are some aspects to be improved and major revision performed before the acceptance. See below the comments:

  1. Line 8.

“HA-targeting allows carbon nanomaterial (CNM)-conjugated carriers” the sentence is confusing, CNM are conjugated to some carriers?? The authors should rewrite it.

In the Abstract section line 18 “ CNM-based nanocomposite” is mentioned, while in title the term is HA- conjugated CNM. It must remove one of them to be consistenst in whole manuscript.

.

  1. Line 30 “HA is a critical component…” Why critical?
  2. Line 32. The high zeta potential confirms its highly hydrophilic properties, enabling it to act as a targeting agent for nanocomposite drug delivery systems”. The sentence is not true, the zeta potential and the hydrophilicity does not ensure targeting properties. HA targeting properties should be adequately explained.
  3. Line 33 The authors mentioned a composite, but do not provide any details on it.
  4. The whole paragraph from lines 58 to 61 is very confusing. The authors must rewrite it to make it intelligible.
  5. Line 165. Phospholipids are substances, not moieties. Also, in the next phrase, the substances that authors mentioned are not relevant “for the biomedical application of hyaluronic acid”. The phrase should be rewritten.

  1. Line 173. “HA can be linked to drugs or drug carriers and might improve the half-life of a drug [57].” The sentence should be continued with some details, examples.

  1. In Section 2, entitled “Carbon nanomaterial conjugated with hyaluronic acid for drug delivery”, the next paragraph suggests that the discussion focus on “how effective the overall nanocomposite, consisting of HA and a specific CNM, was at removing tumour”. The therapeutic efficacy (tumor reduction) of the drug delivery system is mainly due to the active principle encapsulated in the carrier. The HA effect should be highlighted in the context in which efficiency increases due to targeting, but for a carrier with and without targeting.

  1. Line 243 “A few papers showed very promising results from the perspective of reducing tu-mour growth” The term “few” should be explained, i.e there are few papers reporting tumor reduction because most of the papers are reporting in vitro tests (cellular viability) as efficiency evaluation or there are few papers because others do not report promising results.

  1. Line 292 “Cumulative release profiles of CNTs” the section seems to refer to release profile of some anticancer drugs encapsulated in CNTs, the authors should modify the title

  1. Section 2.2.1 and 2.2.2 are devoted to the use of GO in chemo and photoablation therapies, not to the therapies themselves, titles could be improved in this respect.

  1. Section 2.2.5 addresses 3 very different topics, that cannot be brought together in a single chapter and the section should be reorganized. Also the 3 subjects are treated in an extremely elliptical way and the section is disproportionately short. This part, if it is considered important should be extended or, otherwise removed.

  1. In Conclusion section critical aspects should be addressed and perspective on further research in this topic should be added.

Author Response

Dear reviewer,

We thank you for the stimulating and insightful comments, which we believe have served to strengthen the quality of the manuscript.

In response to your comments and recommendations, we have revised the manuscript accordingly and we are resubmitting it for your further consideration. 

Enclosed is a list of point-by-point responses to your comments. All the changes made in the manuscript are highlighted in yellow. 

Reviewer 1

Comment 1: Line 8. “HA-targeting allows carbon nanomaterial (CNM)-conjugated carriers” the sentence is confusing, CNM are conjugated to some carriers?? The authors should rewrite it. In the Abstract section line 18 “ CNM-based nanocomposite” is mentioned, while in title the term is HA- conjugated CNM. It must remove one of them to be consistent in whole manuscript.

Answer 1: We would like to thank you for the comment. To remove the confusion, the word conjugated was removed and it was then referred to as a hybrid system as opposed to a conjugated carrier.

Comment 2: Line 30 “HA is a critical component…” Why critical?

Answer 2: The critical need of HA has been explained in the manuscript for its function in the extracellular matrix and body fluid.

Comment 3: Line 32. The high zeta potential confirms its highly hydrophilic properties, enabling it to act as a targeting agent for nanocomposite drug delivery systems”. The sentence is not true, the zeta potential and the hydrophilicity does not ensure targeting properties. HA targeting properties should be adequately explained.

Answer 3: The section has been improved by detailing that the targeting ability is not a direct consequence of the zeta potential and an example of the zeta potential increase is included.

Comment 4: Line 33 The authors mentioned a composite, but do not provide any details on it.

Answer 4: The name of the composite has now been included.

Comment 5: The whole paragraph from lines 58 to 61 is very confusing. The authors must rewrite it to make it intelligible.

Answer 5: . The paragraph has been re-written to give a better description of the process shown in Figure 2.

Comment 6: Line 165. Phospholipids are substances, not moieties. Also, in the next phrase, the substances that authors mentioned are not relevant “for the biomedical application of hyaluronic acid”. The phrase should be rewritten.

 Answer 6: This technical term has been adjusted for every time it appears.

Comment 7: Line 173. “HA can be linked to drugs or drug carriers and might improve the half-life of a drug [57].” The sentence should be continued with some details, examples.

 Answer 7: The example of Insulin is given with the relevant papers referenced.

Comment 8: In Section 2, entitled “Carbon nanomaterial conjugated with hyaluronic acid for drug delivery”, the next paragraph suggests that the discussion focus on “how effective the overall nanocomposite, consisting of HA and a specific CNM, was at removing tumour”. The therapeutic efficacy (tumor reduction) of the drug delivery system is mainly due to the active principle encapsulated in the carrier. The HA effect should be highlighted in the context in which efficiency increases due to targeting, but for a carrier with and without targeting.

 Answer 8: Changes have been made to be more inclusive of the information given within the following paragraphs and given examples of HA’s importance to each section. 

Comment 9: Line 243 “A few papers showed very promising results from the perspective of reducing tumour growth” The term “few” should be explained, i.e there are few papers reporting tumor reduction because most of the papers are reporting in vitro tests (cellular viability) as efficiency evaluation or there are few papers because others do not report promising results.

 Answer 9: We included a sentence stating the importance of tumour volume reduction profiles.

Comment 10: Line 292 “Cumulative release profiles of CNTs” the section seems to refer to release profile of some anticancer drugs encapsulated in CNTs, the authors should modify the title

 Answer 10: The title has been modified accordingly.

Comment 11: Section 2.2.1 and 2.2.2 are devoted to the use of GO in chemo and photoablation therapies, not to the therapies themselves, titles could be improved in this respect.

 Answer 11: Titles have been improved to give a more specific name to the section.

Comment 12: Section 2.2.5 addresses 3 very different topics, that cannot be brought together in a single chapter and the section should be reorganized. Also the 3 subjects are treated in an extremely elliptical way and the section is disproportionately short. This part, if it is considered important should be extended or, otherwise removed.

 Answer 12: We decided to remove this section.

Comment 13: In Conclusion section critical aspects should be addressed and perspective on further research in this topic should be added.

Answer 13: Additional critical aspects have been included on why HA conjugated carbon nanomaterial delivery system are possible for further research. 

Reviewer 2 Report

The manuscript by Kearns and co-authors describes an extensive literature review on carbon nanomaterials functionalized with hyaluronic acid for tumor management. Important basic characteristics of the delivery systems developed and of the hyaluronic acid mechanism of cellular interaction are exploited. A brief description was given to most commonly used anticancer agents, and then the most fruitful section of the manuscript. Very well organized and detailed, with good selection of examples. Fewer data is available for graphene, yet the review approaches the most common carbon nanomaterials used as delivery system. Focus on phototherapy is highlighted.

The poorer section is the conclusion, where the authors seem "tired". This would be a great opportunity to show the readers the use of this review and the carbon nanomaterials as opposed to others (lipid or polymeric nanoparticles, key players in photochemotherapy). Besides a brief future perspective, recommendations could also improve the manuscript in this section. With this in mind, the revised version could be considered for publication.

Author Response

We would like to thank the reviewer for the positive feedback and constructive and valuable comment. We have revised the conclusion section accordingly and we are resubmitting our revised manuscript for your further consideration. All the changes made in the manuscript are highlighted in yellow. 

Round 2

Reviewer 1 Report

The manuscript could be published in the revised form provided by authors.